# An Enhanced Quantum K-Nearest Neighbor Classification Algorithm Based on Polar Distance

**DOI:** 10.3390/e25010127

**Published:** 2023-01-08

**Authors:** Congcong Feng, Bo Zhao, Xin Zhou, Xiaodong Ding, Zheng Shan

**Affiliations:** 1School of Cyber Science and Engineering, Zhengzhou University, Zhengzhou 450002, China; 2State Key Laboratory of Mathematical Engineering and Advanced Computing, Zhengzhou 450001, China; 3Songshan Laboratory, Zhengzhou 450001, China

**Keywords:** quantum computation, quantum machine learning, K-nearest neighbor algorithm, quantum K-nearest neighbor algorithm

## Abstract

The K-nearest neighbor (KNN) algorithm is one of the most extensively used classification algorithms, while its high time complexity limits its performance in the era of big data. The quantum K-nearest neighbor (QKNN) algorithm can handle the above problem with satisfactory efficiency; however, its accuracy is sacrificed when directly applying the traditional similarity measure based on Euclidean distance. Inspired by the Polar coordinate system and the quantum property, this work proposes a new similarity measure to replace the Euclidean distance, which is defined as Polar distance. Polar distance considers both angular and module length information, introducing a weight parameter adjusted to the specific application data. To validate the efficiency of Polar distance, we conducted various experiments using several typical datasets. For the conventional KNN algorithm, the accuracy performance is comparable when using Polar distance for similarity measurement, while for the QKNN algorithm, it significantly outperforms the Euclidean distance in terms of classification accuracy. Furthermore, the Polar distance shows scalability and robustness superior to the Euclidean distance, providing an opportunity for the large-scale application of QKNN in practice.

## 1. Introduction

Machine learning has made remarkable achievements in various artificial intelligence applications, such as object detection [1,2,3,4], image classification [5,6,7,8], and natural language processing [9,10,11]. However, in the era of big data, we are facing the problem of rapid growth in the amount and type of data. We urgently need to find more high-efficiency computing methods. The quantum system with natural parallelism looks like a good choice. With the in-depth study of quantum technology, many quantum algorithms showing quantum superiority have been proposed [12,13,14,15]. Researchers found that quantum and machine learning algorithms can be combined to improve the performance of the algorithm. The concept of quantum machine learning was born [16,17]. Many quantum machine learning algorithms [18,19,20,21,22] are significantly better than their classical counterparts. In this context, a KNN algorithm with a simple idea but high time complexity has attracted the interest of researchers. It requires little to no prior knowledge when classifying [23]. Similarity calculation and K-nearest neighbors search are two important parts of KNN. In recent years, many quantum methods for these two processes have been proposed. In 2001, Harry Buhrman et al. proposed the swap test quantum circuit for calculating the cosine distance of two vectors [24]. In 2013, Lloyd et al. proposed a quantum Euclidean distance estimator based on the swap test circuit [25]. Based on this, Wiebe et al. proposed a quantum nearest neighbor algorithm [26] in 2014 and used Dürr and Høyer’s algorithm for finding the minimum value in a database [27] to find the nearest neighbor. For non-numerical data, a quantum K-nearest neighbor algorithm based on Hamming distance is proposed [28,29].

The similarity measure, which affects the accuracy of the algorithm classification, lies at the heart of the K-nearest neighbor algorithm [30]. A similarity measure is used to measure how similar two things are [31]. To date, many similarity measures have been proposed, such as Euclidean distance, cosine distance, Hamming distance, and so on. However, there is no one similarity distance measure that can best solve all problems [31]. Choosing an appropriate similarity measure will significantly improve the K-nearest neighbor algorithm’s classification accuracy. The Euclidean distance is the most frequently applied similarity measure. However, the result of the quantum Euclidean distance estimator has poor stability, and there is a significant difference with the actual result [32]. Therefore, we need to find a new similarity measure to replace the use of Euclidean distance in QKNN.

In machine learning, a sample is usually regarded as a vector with both magnitude and direction. Inspired by this, in addition to using Cartesian coordinates, we can also use Polar coordinates to represent a sample. So, we propose a new similarity measure that we call Polar distance, which considers both angular and module length information. The cosine theorem shows that the Euclidean distance is a combination of angular and module length information. The Polar distance introduces an adjustable parameter to adjust the ratio of angular and module length information according to the specific application. Then, we propose a quantum circuit to calculate the Polar distance. The frame diagram of the quantum part of the QKNN algorithm is shown in Figure 1. We optimize Figure 1b in this work. The following is a list of our major contributions:(1)We propose a new similarity measure, called Polar distance, which integrates both angular and module length information and combines the two proportionally according to practical applications. Its classification accuracy in KNN is comparable to that of Euclidean distance;(2)We design a quantum circuit to calculate the Polar distance. Compared with the quantum Euclidean distance estimator, it can directly obtain the desired results and has less difference with the classical results;(3)We carry out KNN and QKNN(quantum simulation) experiments on different datasets. The KNN’s experimental results show that Polar distance is comparable to Euclidean distance in classification accuracy. The QKNN’s experimental results show that Polar distance is better than Euclidean distance in classification accuracy.

**Figure 1 entropy-25-00127-f001:**
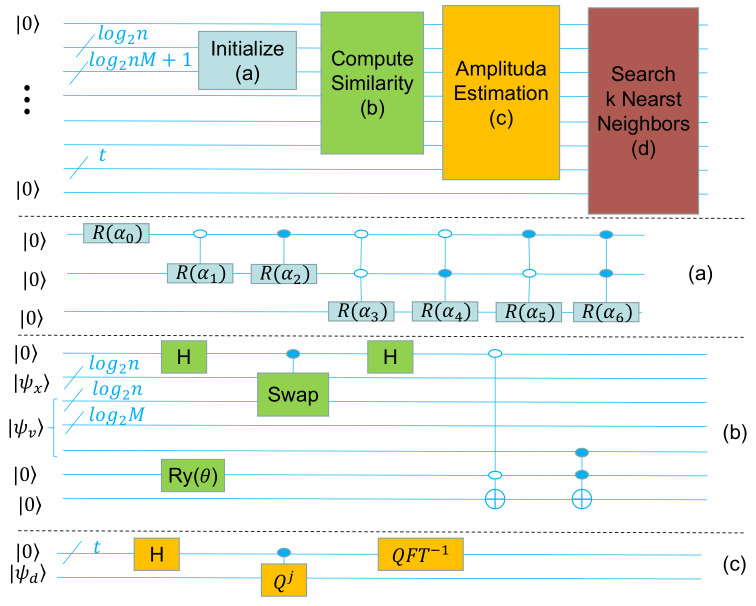
Frame diagram of quantum part of QKNN algorithm: (**a**) initialize any quantum state circuit, where α is the calculated parameter based on the quantum state vector representation; (**b**) quantum circuit for calculating Polar distance, where $|\psi_{x}\rangle$ represents the test sample and $|\psi_{v}\rangle (|\psi_{v}\rangle =|v_{i}\rangle |i\rangle |d_{r_{i}}\rangle )$ represents the entangled superposition state of the module length similarity and the training set; (**c**) amplitude estimation circuit diagram.

## 2. Materials and Methods

### 2.1. KNN

K-nearest neighbor algorithm is a supervised machine learning algorithm [23]. Its general idea is: if most of the K-nearest samples to a sample belong to a given category in the feature space, then the sample belongs to that category as well. The whole process of KNN is shown in Algorithm 1. Its main steps are as follows: first, calculate the similarity between the test sample and all training samples; then find the *k* training samples that are most like the test sample; finally, according to the category of *k* training samples, the category of the test sample is determined according to the principle that the minority obeys the majority. For example, as shown in Figure 2, the question mark represents the unknown test sample, and the red circle and blue cross denote two categories of training samples. At *k* = 1, the category of the question mark is consistent with the red circle category. When *k* = 5, the category of the question mark is consistent with the blue cross category. Obviously, the classification results will be affected by *k* value. Furthermore, similarity measure is another factor influencing the results of classification.
**Algorithm 1** KNN**Input:** A test sample and some training samples**Output:** The test sample’s category
 1: **for** number of training samples do **do** 2:  calculate the similarity between the test sample and a training sample
 3: **end for**
 4: find the *k* training samples that are most like the test sample
 5: determine test sample’s category


### 2.2. QKNN

The quantum K-nearest neighbor algorithm is consistent with the overall idea of the classical K-nearest neighbor algorithm. Quantum K-nearest neighbor algorithm quantizes the part of K-nearest neighbor algorithm with high time complexity. It uses the natural parallelism of quantum computing to reduce the time complexity of the algorithm. As shown in Figure 1, the quantum part of the quantum K-nearest neighbor proposed in this paper consists of four parts: Initialize, Compute Similarity, Amplitude Estimation, and Search K-Nearest Neighbors. Finally, the test sample category is determined by the classical method. We describe these five parts in detail later.

#### 2.2.1. Initialize

In order to process classical data using the quantum system, we need to encode classical data into quantum state. At present, there are many methods to encode classical data into quantum states [33,34,35]. The coding methods can be divided into two categories: using the amplitude of the quantum state to encode information and directly using the quantum state to encode information. Amplitude coding is one of the common coding methods in quantum machine learning algorithms [36]. In this paper, we also use amplitude coding. Its main idea is to use the amplitude of quantum states to represent classical data. In order to represent the classical vector by amplitude, we must first normalize the vector so that the vector module length is 1. After that, we should ensure that the dimension of the vector is $2^{n}$, and n is the number of qubits required to encode the vector. When the vector does not meet this condition, it is completed by supplementing 0. Take vector $\vec{a}$ as an example.
(1)$$\vec{a}=\left(a_{0},a_{1},\ldots ,a_{2^{n}-1}\right)$$Its quantum representation is as follows:(2)$$|\psi_{a}\rangle =\sum_{i=0}^{2^{n}-1}\frac{a_{i}}{\sqrt{|a_{0}{|}^{2}+|a_{1}{|}^{2}+\ldots +{\left|a_{2^{n}-1}\right|}^{2}}}|i\rangle$$Then, we need to initialize the register of *n* qubits as $|\psi_{a}\rangle$. The initial state is |0…0〉, we start from the high position, and the quantum circuit is shown in Figure 1a. The Ry represents single-qubit rotation about the Y-axis, the solid point represents 1 control, and the hollow point represents 0 control.

#### 2.2.2. Compute Similarity

Inspired by Polar coordinates, we propose a parametric similarity measure that combines cosine similarity and module length similarity, which we call Polar distance.
(3)$$d=d_{c}\cdot \left(1-\omega \right)+d_{r}\cdot \omega$$Among them, $d_{c}$ and $d_{r}$ represent cosine similarity and module length similarity, respectively, ω represents an adjustable parameter, and the value range of the three is [0, 1]. As *d* increases, the similarity between the two samples is stronger, otherwise this similarity becomes weaker. We can adjust the value of ω to improve the classification accuracy according to the actual application. The formula for calculating the cosine similarity is as follows (θ represents the angle between two vectors):(4)$$d_{c}=0.5\cdot \left(1+cos^{2}\theta \right)$$The formula for calculating the module length similarity is as follows:(5)$$d_{r}=1-\left|r_{x}-r_{v}\right|$$
where $r_{x}$ refers to the module length of the test sample and $r_{v}$ represents the module length of the training sample. The closer the $d_{r}$ value is to 1, the greater the similarity between the two samples, and the closer the $d_{r}$ value is to 0, the smaller the similarity between the two samples.

As shown in Figure 1b, the similarity calculation is divided into two steps: first, the swap test circuit is applied to calculate the cosine similarity [24], and then the weighted summation circuit proposed by us is used to realize the weighted summation of the cosine similarity and the module length similarity.

Next, we take the calculation of the similarity between two samples *x* and *v* as an example to introduce the calculation process in detail. The initial state of the quantum system is:(6)$$|s_{0}\rangle =|0\rangle |x\rangle |v\rangle (\sqrt{1-d_{r}}|0\rangle +\sqrt{d_{r}}\left|1\rangle \right)(\sqrt{1-\omega }|0\rangle +\sqrt{\omega }|1\rangle )|0\rangle$$First of all, after a Hadamard gate is utilized, the state of the quantum system becomes:(7)$$|s_{1}\rangle =|+\rangle |x\rangle |v\rangle (\sqrt{1-d_{r}}|0\rangle +\sqrt{d_{r}}\left|1\rangle \right)(\sqrt{1-\omega }|0\rangle +\sqrt{\omega }|1\rangle )|0\rangle$$Then, after the usage of CSWAP gate, the state of the quantum system transforms into:(8)$$|s_{2}\rangle =\frac{1}{\sqrt{2}}(|0\rangle |x\rangle |v\rangle +\left|1\rangle |v\rangle |x\rangle \right)(\sqrt{1-d_{r}}|0\rangle +\sqrt{d_{r}}\left|1\rangle \right)(\sqrt{1-\omega }|0\rangle +\sqrt{\omega }|1\rangle )|0\rangle$$At the third stage, another Hadamard gate is applied. The state of the quantum system is given as:(9)$$\begin{array}{cc} |s_{3}\rangle & =\frac{1}{2}(|0\rangle |x\rangle |v\rangle +|1\rangle |x\rangle |v\rangle +|0\rangle |v\rangle |x\rangle -\left|1\rangle |v\rangle |x\rangle \right)\\ \; & (\sqrt{1-d_{r}}|0\rangle +\sqrt{d_{r}}\left|1\rangle \right)(\sqrt{1-\omega }|0\rangle +\sqrt{\omega }|1\rangle )|0\rangle \end{array}$$At the fourth stage, the extended general Toffoli gate is applied, and the state of the quantum system reads:(10)$$\begin{array}{cc} |s_{4}\rangle & =\frac{1}{2}|0\rangle |x\rangle |v\rangle (\sqrt{1-d_{r}}|0\rangle +\sqrt{d_{r}}\left|1\rangle \right)(\sqrt{1-\omega }|0\rangle |1\rangle +\sqrt{\omega }\left|1\rangle |0\rangle \right)\\ \; & +\frac{1}{2}|1\rangle |x\rangle |v\rangle (\sqrt{1-d_{r}}|0\rangle +\sqrt{d_{r}}\left|1\rangle \right)(\sqrt{1-\omega }|0\rangle |0\rangle +\sqrt{\omega }\left|1\rangle |0\rangle \right)\\ \; & +\frac{1}{2}|0\rangle |v\rangle |x\rangle (\sqrt{1-d_{r}}|0\rangle +\sqrt{d_{r}}\left|1\rangle \right)(\sqrt{1-\omega }|0\rangle |1\rangle +\sqrt{\omega }\left|1\rangle |0\rangle \right)\\ \; & +\frac{1}{2}|1\rangle |v\rangle |x\rangle (\sqrt{1-d_{r}}|0\rangle +\sqrt{d_{r}}\left|1\rangle \right)(\sqrt{1-\omega }|0\rangle |0\rangle +\sqrt{\omega }\left|1\rangle |0\rangle \right) \end{array}$$At the fifth stage, the Toffoli gate is applied, and the state of the quantum system becomes:(11)$$\begin{array}{cc} |s_{5}\rangle & =\frac{1}{2}|0\rangle |x\rangle |v\rangle \sqrt{1-d_{r}}|0\rangle \sqrt{1-\omega }|0\rangle |1\rangle +\frac{1}{2}|0\rangle |x\rangle |v\rangle \sqrt{1-d_{r}}|0\rangle \sqrt{\omega }|1\rangle |0\rangle \\ \; & +\frac{1}{2}|0\rangle |x\rangle |v\rangle \sqrt{d_{r}}|1\rangle \sqrt{1-\omega }|0\rangle |1\rangle +\frac{1}{2}|0\rangle |x\rangle |v\rangle \sqrt{d_{r}}|1\rangle \sqrt{\omega }|1\rangle |1\rangle \\ \; & +\frac{1}{2}|1\rangle |x\rangle |v\rangle \sqrt{1-d_{r}}|0\rangle \sqrt{1-\omega }|0\rangle |0\rangle +\frac{1}{2}|1\rangle |x\rangle |v\rangle \sqrt{1-d_{r}}|0\rangle \sqrt{\omega }|1\rangle |0\rangle \\ \; & +\frac{1}{2}|1\rangle |x\rangle |v\rangle \sqrt{d_{r}}|1\rangle \sqrt{1-\omega }|0\rangle |0\rangle +\frac{1}{2}|1\rangle |x\rangle |v\rangle \sqrt{d_{r}}|1\rangle \sqrt{\omega }|1\rangle |1\rangle \\ \; & +\frac{1}{2}|0\rangle |v\rangle |x\rangle \sqrt{1-d_{r}}|0\rangle \sqrt{1-\omega }|0\rangle |1\rangle +\frac{1}{2}|0\rangle |v\rangle |x\rangle \sqrt{1-d_{r}}|0\rangle \sqrt{\omega }|1\rangle |0\rangle \\ \; & +\frac{1}{2}|0\rangle |v\rangle |x\rangle \sqrt{d_{r}}|1\rangle \sqrt{1-\omega }|0\rangle |1\rangle +\frac{1}{2}|0\rangle |v\rangle |x\rangle \sqrt{d_{r}}|1\rangle \sqrt{\omega }|1\rangle |1\rangle \\ \; & -\frac{1}{2}|1\rangle |v\rangle |x\rangle \sqrt{1-d_{r}}|0\rangle \sqrt{1-\omega }|0\rangle |0\rangle -\frac{1}{2}|1\rangle |v\rangle |x\rangle \sqrt{1-d_{r}}|0\rangle \sqrt{\omega }|1\rangle |0\rangle \\ \; & -\frac{1}{2}|1\rangle |v\rangle |x\rangle \sqrt{d_{r}}|1\rangle \sqrt{1-\omega }|0\rangle |0\rangle -\frac{1}{2}|1\rangle |v\rangle |x\rangle \sqrt{d_{r}}|1\rangle \sqrt{\omega }|1\rangle |1\rangle \end{array}$$Finally, the last qubit is measured. The probability of getting state |1〉, which measures the last qubit with the basis state |1〉, is given by:(12)$$p(|1\rangle )=\left(\frac{1}{2}+\frac{1}{2}{\left|\langle x|v\rangle \right|}^{2}\right)\left(1-\omega \right)+d_{r}\omega$$

#### 2.2.3. Amplitude Estimation

There are two methods to obtain the results of similarity calculation of quantum circuits. The first method is to obtain statistical results through multiple measurements. The disadvantage of this method is that it cannot determine the accuracy of the results. The other method is to use an amplitude estimation algorithm [37]. Amplitude estimation can control the accuracy of the results by adjusting the number of qubits. In addition, the use of amplitude estimation is more convenient for subsequent quantum steps. So, we use amplitude estimation. In this article, we only introduce the usage of amplitude estimation. For more details, please refer to [37]. The function of amplitude estimation is to calculate the value of *a* in Equation (Equation 13). The circuit of amplitude estimation is shown in Figure 1c. The first step is to initialize the two registers with status |0〉A|0〉. The second step is to apply QFT to the first register. The third step is to apply a controlled $Q^{j}\left(Q=-AS_{0}A^{-1}S_{X}\right)$. The fourth step is to apply $QFT^{-1}$ to the first register. The fifth step is to measure the first register and denote the outcome |y〉. Finally, calculate the amplitude $a=sin(\pi \frac{y}{2^{t}})$. We can control the precision of the result by adjusting the number of qubits *t*. The higher the value of *t*, the higher the precision of the result. On the contrary, the lower the precision. The functions of unitary operators $A,S_{0}$ and $S_{X}$ are as follows:(13)$$A|0\rangle =a|\psi \rangle +\sqrt{1-a^{2}}|\psi_{⊥}\rangle$$
(14)$$S_{0}=I-2|0\rangle \langle 0|$$
(15)$$S_{X}=I-2|\psi \rangle \langle \psi |$$In order to apply the quantum algorithm for finding the K-nearest neighbors, we need to make the amplitude estimation step reversible. Wiebe et al. call this form of amplitude estimation coherent amplitude estimation [26]. This results in a state that is, up to local isometries, approximately
(16)$$\frac{1}{\sqrt{M}}\sum_{j=0}^{M}|j\rangle ||x-v_{j}|\rangle$$

#### 2.2.4. Search K-Nearest Neighbors

Searching K-nearest neighbors is a part of a KNN with high time complexity. The appearance of the Grover algorithm opens up a new idea for the unordered search problem [12]. Dürr proposed a quantum algorithm [38] for finding *k*-minimum values in 2004. Miyamoto proposed a quantum algorithm to find *K*-minimum values with another idea in 2019 [39]. Both their algorithms are capable of finding *k*-minimum values in *M* data with a time complexity of $O(\sqrt{kM})$. Miyamoto’s algorithm is simpler and easier to implement. Here, we will present their method. He introduced a parameter *t*, which is used to find out *k* values that are less than *t*. The quantum algorithm for finding k-Minima is summarized as follows:(1)Apply algorithm [27] for finding minimum and record the last *k* indexes of finding minimum algorithm process;(2)Use binary search to find the minimum algorithm record of the threshold index *t* that meets the condition that the number less than *t* is close to *k*. The quantum counting algorithm is used to determine whether the condition is satisfied;(3)Apply Grover algorithm to search *k* values that are less than *t*.

According to [29], the time complexity of the first step is $O(\sqrt{M})$. The second step combines the quantum counting algorithm with the classical binary search. Its time complexity is $O(\sqrt{M}logk)$. Finally, the time complexity of searching K indexes is $O(\sqrt{kM})$. To sum up, the overall time complexity of the algorithm is given as:(17)$$\left(\sqrt{M}\right)+O\left(\sqrt{M}logk\right)+O\left(\sqrt{kM}\right)=O\left(\sqrt{kM}\right)$$

#### 2.2.5. Determine Category

Finally, we determine the category of the test sample according to the *k* most similar training samples. Suppose the number of each category in the *k* most similar training samples is $k_{i}$. The category of test samples is consistent with the index of $max(k_{i})$. However, in practical application, $max\left(k_{i}\right)=k_{a}=k_{b}\left(a\neq b\right)$ may take place, which makes it impossible to determine the type of the test sample. In this paper, once the onset of above issue, we make k=k+1 until we can determine the category of the test sample.

## 3. Results

In this section, we first demonstrate theoretically that the Polar distance can be used as a measure of sample similarity. We then compare the performance of Polar distance and Euclidean distance in KNN on the Iris, Wine, Liver, and Overflow Vulnerability datasets. Finally, we compared the performance of Polar distance and Euclidean distance in QKNN on the same dataset. The accuracy of all experiments is the average of 30 10-fold cross-validations.

### 3.1. A New Similarity Distance Measure

Similarity measure is a metric for comparing the similarity of two samples. When comparing two samples, distance is usually used to determine their similarity. In this paper, we proposed a new similarity distance measure called Polar distance that considers both the information of angle and module length by combining them into a weighted value. In general, distance should meet the following three properties: non-negativity, symmetry, and trigonometric inequality. Derived by cosine similarity, the angle can be used as an index to measure the similarity. Here, we prove that module length can be used as an indicator for similarity measurement from the three properties of distance above. In this work, we define the module length distance of two samples A and B as:(18)$$|r_{A}-r_{B}|$$The first is non-negativity and symmetry. Obviously,
(19)$$|r_{A}-r_{B}|\geq 0$$
(20)$$|r_{A}-r_{B}\left|=\right|r_{B}-r_{A}|$$Finally, it is proved that it satisfies the trigonometric inequality. Considering any three samples, such as A, B, and C, it is necessary to prove
(21)$$|r_{A}-r_{B}\left|+\right|r_{A}-r_{C}\left|\geq \right|r_{B}-r_{A}+r_{A}-r_{C}\left|=\right|r_{B}-r_{C}|$$Obviously, it satisfies the trigonometric inequality. Module length distance can be used as an indicator to measure similarity. In order to combine the angle and module length as indicators to measure similarity, we define the new similarity measurement method as the following form:(22)$$d=0.5\cdot \left(1+cos^{2}\theta \right)\cdot \left(1-\omega \right)+(1-|r_{A}-r_{B}\left|\right)\cdot \omega$$
where module length $r_{A}$ and $r_{B}$ are scaled so that their value range is [0, 1]. The ω values in this paper were determined using cross-validation. Specifically, the value of *k* under Euclidean distance is first determined using cross-validation. The value of *k* is then held constant, and we determine the parameter ω for the Polar distance using the cross-validation method.

### 3.2. Polar Distance and Euclidean Distance in KNN

To verify that the Polar distance can replace the Euclidean distance in KNN, we first tested the classification accuracy of two similarity distance measures in KNN under different datasets. Iris and Wine are datasets with three classes. Overflow Vulnerability and Liver are datasets with two classes. There is not much difference in the classification accuracy of the two similarity distance measures, as shown in Figure 3, Figure 4, Figure 5 and Figure 6. Table 1’s KNN column shows that the difference between the two similarity distance measures is still small in the best shape of the classification accuracy. Therefore, we consider that the two similarity distance measures are approximately equivalent in KNN.

### 3.3. Polar Distance and Euclidean Distance in QKNN

To validate that the Polar distance can replace the Euclidean distance in QKNN, we make similar experiments as above. As shown in Figure 3, Figure 4, Figure 5 and Figure 6, we can easily see that the gap between the Polar distance and quantum Polar distance is significantly smaller than the gap between the Euclidean distance and quantum Euclidean distance. From the results in the QKNN column of Table 1, the Polar distance we proposed is better than the Euclidean distance in classification accuracy. For the dataset of Iris, the accuracy of Polar distance is 95.82%, achieving a 9.8% accuracy gain against Euclidean distance. For the dataset of Wine, the accuracy of Polar distance is 95.86%, achieving a 1.65% accuracy gain against Euclidean distance. For the dataset of Liver, the accuracy of Polar distance is 89.19%, achieving a 2.13% accuracy gain against Euclidean distance. For the dataset of Overflow Vulnerability, the accuracy of Polar distance is 63.42%, achieving a 16.09% accuracy gain against Euclidean distance. It is well known that there are deviations between quantum results and theoretical values. Although our QKNN experiments are performed by a quantum simulator, this deviation still exists due to Monte Carlo sampling. So why is the deviation from the quantum Euclidean distance greater? This starts with the calculation of the quantum Euclidean distance [32]. The formula for calculating the quantum Euclidean distance is as follows:(23)$$d=\sqrt{2*\left(r_{1}^{2}+r_{2}^{2}\right)*\left(2*p(|0\rangle \right)-1)}$$Assume that the error of the quantum measurement result is δ. Obviously the error of the quantum Polar distance is δ. The errors in the quantum Euclidean data are as follows:(24)$$\frac{\left|\Delta d\right|}{d}=\frac{2\delta p}{\sqrt{2p-1}*\left(\sqrt{2p-1}+\sqrt{2*(1\pm \delta )p-1}\right)}$$The error of the quantum Euclidean distance is $\frac{\left|\Delta d\right|}{d}=\frac{2p}{\sqrt{2p-1}*\left(\sqrt{2p-1}+\sqrt{2*(1\pm \delta )p-1}\right)}$ (the value of the equation is 1 to +∞ since p∈[0.5,1]) times that of the quantum Polar distance. The classical part of the quantum Euclidean distance estimator amplifies the error in the quantum part, so the results differ significantly from the true results. This leads to less satisfactory results for the quantum K-nearest neighbor (QKNN) algorithm based on Euclidean distance.

## 4. Discussion

In this paper, we proposed a new similarity distance measure to replace the Euclidean distance for use in QKNN. We call it Polar distance. From the experimental results, the Polar distance can achieve the following results in terms of classification accuracy:(1)The Polar and Euclidean distances are comparable in KNN;(2)The Polar distances are comparable in KNN and QKNN;(3)The Polar distances perform significantly better than the Euclidean distances in QKNN.

However, the disadvantage of the Polar distance is also obvious, namely the introduction of a new parameter ω. This not only increases the computational complexity but also makes Polar distance only applicable in supervised machine learning algorithms. We can try to find the right value of ω quickly by using gradient descent. Experiments have shown that the value of ω is not the same under different datasets. We can also delve into the relationship between the value of ω and the distribution of samples in the dataset to address the above problems. This is worthy of further study.

## Figures and Tables

**Figure 2 entropy-25-00127-f002:**
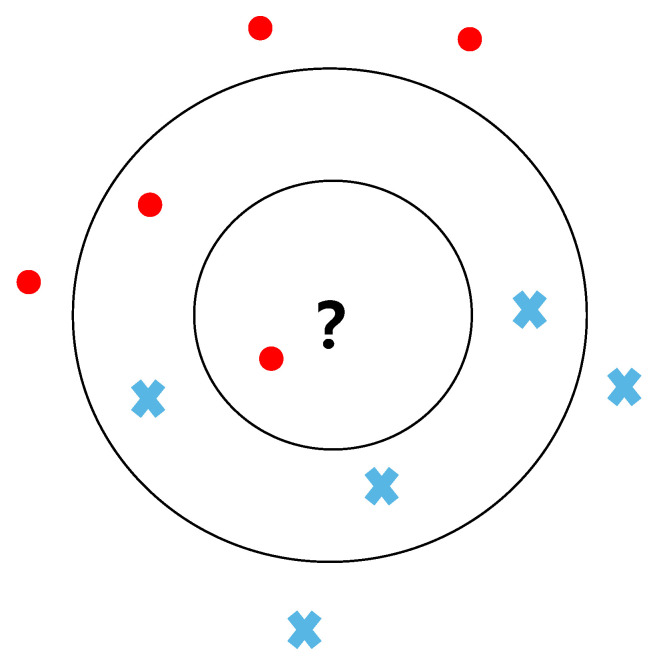
Schematic of the KNN algorithm.

**Figure 3 entropy-25-00127-f003:**
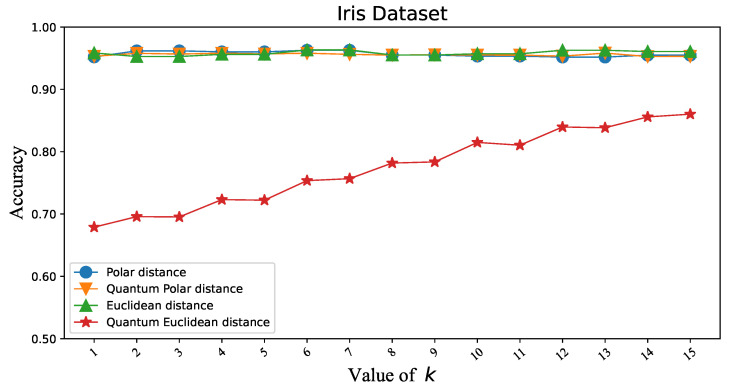
Classification accuracy corresponding to Polar distance and Euclidean distance in KNN and QKNN on the Iris dataset.

**Figure 4 entropy-25-00127-f004:**
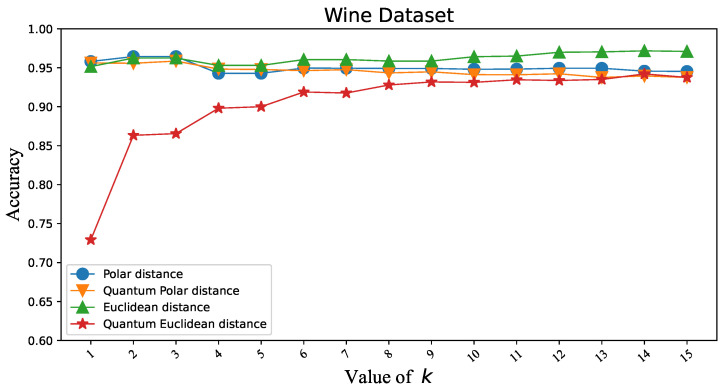
Classification accuracy corresponding to Polar distance and Euclidean distance in KNN and QKNN on the Wine dataset.

**Figure 5 entropy-25-00127-f005:**
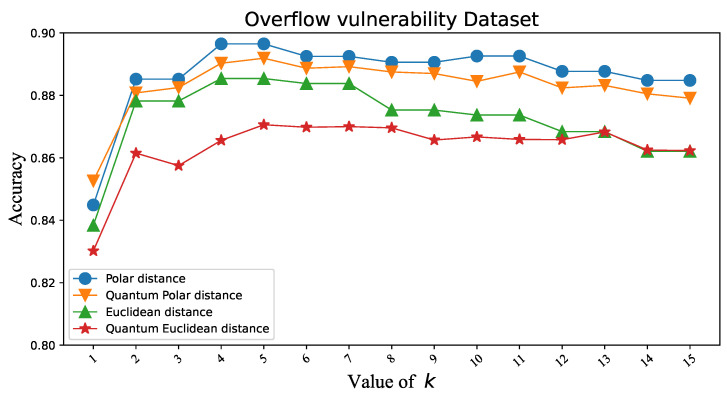
Classification accuracy corresponding to Polar distance and Euclidean distance in KNN and QKNN on the Overflow Vulnerability dataset.

**Figure 6 entropy-25-00127-f006:**
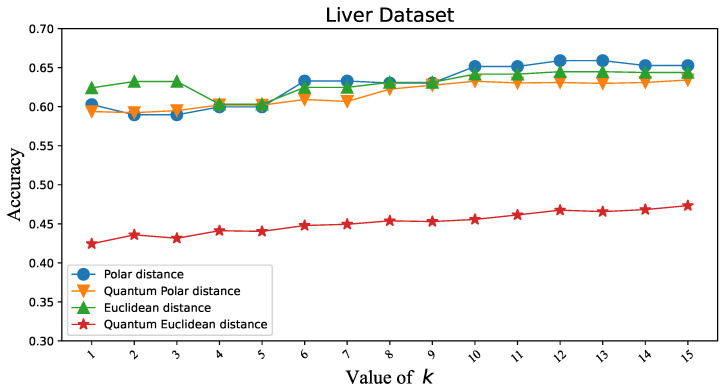
Classification accuracy corresponding to Polar distance and Euclidean distance in KNN and QKNN on the Liver dataset.

**Table 1 entropy-25-00127-t001:** Classification accuracy corresponding to Polar distance and Euclidean distance in KNN and QKNN on four datasets.

Datasets	KNN		QKNN	
	Polar Distance	Euclidean Distance	Polar Distance	Euclidean Distance
Iris	96.27%	96.33%	95.82%	86.02%
Wine	96.44%	97.17%	95.86%	94.21%
Overflow	89.65%	88.54%	89.19%	87.06%
Liver	65.90%	64.48%	63.42%	47.33%

## Data Availability

The data that support the findings of this study are available from the corresponding author upon reasonable request.

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
