# Peer review of "An Enhanced Quantum K-Nearest Neighbor Classification Algorithm Based on Polar Distance"

_entropy, 2023, doi:10.3390/e25010127_

Round 1
Reviewer 1 Report
The authors propose a new similarity distance measure "polar distance" for use in quantum k-nearest neighbor (QKNN). They claim that the polar distance can achieve comparable results to the Euclidean distance in traditional k-nearest neighbor (KNN) algorithms, and significantly better results in QKNN algorithms. The polar distance introduces a new free parameter that can be tuned to improve the classification results.
Overall I believe this work is interesting and worthy of consideration, provided that it is presented more clearly.
Some specific points:
* In my opinion the introduction has too many references (1 to 15) that are only loosely connected to the main topic of this work. It might be worth reevaluating the inclusion of some of them.
* There are missing references in lines 34 and 117.
* Figure 2 is mentioned before Figure 1.
* In Equation (2) I believe the squares should be of normed amplitudes
* Cosine similarity / modulo length similarity: These should be defined clearly before use. "Module length" is used later in the text instead of "modulo". Without providing a definition it's not clear which is correct. I suspect it's "modular".
θ is mentioned but is not defined. What is the role of θ? A diagram here would help.
* Equation numbering is not consistent. For example in "the value of a in equation (21)" the equation should be (13) perhaps? Similarly in "Therefore, it is obvious that the inequality (4) ...", etc.
* In my opinion there is no need to prove that these functions define a distance. At least not so analytically.
* I don't think 3.1 section is necessary. It is a repetition of previous sections presented differently and inconsistently. I would suggest this be merged with the background section.
* An algorithmic view (e.g. a list-summary of the steps) would be very helpful.
Author Response
Thank you very much for your valuable suggestions. We have completed the revision of the manuscript. Please read Respond1.docx for details.

Reviewer 2 Report
The manuscript proposes to use a new type of distance paradigm (called polar distance) for the quantum k-nearest neighbour algorithm. The obtained results corroborate the theoretical assumptions of increased performance of polar distance over Euclidean distance for the quantum case.
The manuscript can be improved in terms of grammar, citations and consistency of terminology as suggested below.
34: Missing citation
54: I believe it should have been "modulo length information" instead of "mode length information"
76: I believe you mean "the test sample" by "test the sample"
92: "Its quantum representation is as following". It should be "as follows"
97: "As the increase of d, the similarity between the two samples is stronger, otherwise this similarity becomes weaker". It should be "As d increases, the similarity between the two samples becomes stronger, ..."
106: "Wiebe et al. call this form amplitude estimation coherent amplitude estimation". Please add "of" after "form".
110: Citation missing
112: Please replace "capable to find" with "capable of finding"
117: Citation missing
117: "algorithm for finding minimum"
137: "...the inequality (4) are established" should read "is established"
142: "... and we determine ..."
You have "mode", "module" and "modulo" length throughout the text. Please make it consistent.
As the paper was submitted to the sections "Advances in quantum computing" and "Quantum information", I find it desirable and instructive to the wider audience of the journal to include a paragraph in the introduction section to incorporate a general description of the field of quantum information to make it complete. I suggest to add a paragraph outlining the development in the field to include references to the recent development in quantum information for biology and medicine:
[1] Weng-Long Chang, Ju-Chin Chen, Wen-Yu Chung, Chun-Yuan Hsiao, Renata Wong, Athanasios V Vasilakos. Quantum speedup and mathematical solutions of implementing bio-molecular solutions for the independent set problem on IBM quantum computers. IEEE Transactions on NanoBioscience 20(3): 354-376, 2021. Doi: 10.1109/TNB.2021.3075733
[2] Renata Wong and Weng-Long Chang. Fast Quantum Algorithm for Protein Structure Prediction in Hydrophobic-Hydrophilic Model. Journal of Parallel and Distributed Computing 164: 178-190, 2022. Doi: 10.1016/j.jpdc.2022.03.011
[3] W. -L. Chang, J. -C. Chen, W. -Y. Chung, C. -Y. Hsiao, R. Wong and A. V. Vasilakos . Quantum Speedup for Inferring the Value of Each Bit of a Solution State in Unsorted Databases Using a Bio-Molecular Algorithm on IBM Quantum's Computers. IEEE Transactions on NanoBioscience 21(2): 286-293, 2022. Doi: 10.1109/TNB.2021.3130811
[4] Renata Wong and Weng-Long Chang. Quantum Speedup for Protein Structure Prediction. IEEE Transactions on NanoBioscience 20(3): 323-330, 2021, doi: 10.1109/TNB.2021.3065051.
Author Response
Thank you very much for your valuable suggestions. We have completed the revision of the manuscript. Please read Respond2.docx for details.

Reviewer 3 Report
entropy-2109164
An Enhanced Quantum K-Nearest Neighbor Classification
Algorithm Based on Polar Distance
Congcong Feng et al
The authors study a quantum clssification algorithm for k-nearest neighbor. The authors consider two measures of distances: polar vs Euclidean and show that for classical KNN, there is no significant difference in the performance between the two measures but for the quantum version, there is a significant improvements in using polar distance compared to Euclidean distance. The authors also show that there is no significant difference in performace between KNN and QKNN. Is there a reason for it? If QKNN does not improve over its classical version, why bother? Is there a saving in terms of computational time or memory space? Some estimations of upper bounds (in terms of complexity) might be useful.
Overall, the paper, as it stands needs substantial improvement in its presentation and language. Some figures needs redrawings: in particular, the ordering of the subfigures in Fig 1 leaves much to be desired. In figure 2, the blue cross is called "blue fork". I think it is clearer as "blue cross". Section 2.1 also needs rewording and elucidation.
It is also not obvious what is new in this paper since QKNN has been published before, including one by Basheer et al (Basheer, A., Afham, A., & Goyal, S. K. (2020). Quantum $ k $-nearest neighbors algorithm. arXiv preprint arXiv:2003.09187.) which is not cited in the references. If it is simply a comparison of the effects of different distance measures, the authors will need to elaborate on why they have chosen these two measures. After all, there are tons of valid distances measures and one needs to justify the choices made.
In summary, while I think the paper may eventually be published in some form, it does not carry the oomphs needed for a good paper. I would therefore not recommend publication in Entropy in its current form.
Author Response
Thank you very much for your valuable suggestions. We have completed the revision of the manuscript. Please read Respond3.docx for details.

Round 2
Reviewer 1 Report
The authors have addressed some of the points made. Even though I would prefer a more detailed exposition of the results, the results seem interesting and correct; therefore I would be in favor of acceptance once the authors address the minor point below.
Point 4: In Equation (2) I believe the squares should be of normed amplitudes
Response 4: After careful consideration, we believe that Equation (2) is correct. Equation (2) represents a quantum state that should satisfy .
-
The amplitudes (ai) are in general complex numbers so a2 is not the same as a*a = |a|2 . The denominator of equation (2) needs |a|2 in place of a2 unless a is always a real number.
Author Response
Thank you very much for your suggestion. We have revised the manuscript as suggested. Please see Respond1.docx for details.

Reviewer 3 Report
The authors have addressed some of my concerns. I still find the results incremental, but as the paper is scientifically correct and the results may cast new insights into the quantum k-nearest neighbor classification, I would recommend acceptance once the authors have made the necessary minor changes.
Minor:
In line 34, the authors insert "...D\"{u{rr Hoyer's Finding the Minimum Quantum Algorithm". Perhaps they should just write "D\:{u}rr and Hoyer's algorithm for finding the minimum value in a database".
Figure 1 (b) and (c) should be swapped as most readers read from left to right.
Author Response
Thank you very much for your suggestion. We have revised the manuscript as suggested. Please see Respond3.docx for details.
